# OptiBench: Benchmarking Large Language Models in Optimization Modeling with Equivalence-Detection Evaluation

## Abstract

In operations research (OR), formulating optimization problems in industrial applications is often time-consuming and requires specialized expertise. Recently, large language models (LLMs) have shown remarkable potential to automate this process. However, evaluating the performance of LLMs in optimization modeling remains challenging due to the scarcity of suitable datasets and rigorous evaluation methodologies. To reduce this gap, we introduce OptiBench, a new benchmark designed to assess LLMs' ability to formulate linear programming (LP) and mixed-integer linear programming (MILP) models. OptiBench provides a diverse dataset covering 816 optimization modeling word problems across 16 problem classes and over 80 practical domains. It also adopts a model-data separation format with 2 levels of description abstraction. The dataset exhibits the complexity of real-world optimization problems compared to traditional textbook examples. OptiBench incorporates a new evaluation method based on a modified Weisfeiler-Lehman graph isomorphism test (WL-test) algorithm. We theoretically prove that this method can correctly judge whether two models are equivalent or not, setting a new standard for automatically validating the correctness of optimization modeling. We benchmark various LLMs using OptiBench and observe significant performance differences. GPT-4o by direct prompting achieves 49.39% overall accuracy, outperforming other models and LLM-based agents, including OpenAI o1 (preview and mini). Notably, GPT-4o's performance varies across different problem classes, achieving over 90% accuracy on the knapsack problem class but falling below 5% on the traveling salesman problem class. These findings provide new insights into the strengths and limitations of LLMs in optimization modeling.

## 1 Introduction

Operations Research (OR) is a discipline that employs advanced analytical methods, such as mathematical modeling and optimization techniques, to aid decision-making and problem-solving in complex systems Hillier & Lieberman (2015). OR methods play an important role in various industries such as logistics, manufacturing, finance, and healthcare, optimizing processes and resources to enhance efficiency and productivity Winston (2004). However, the formulation of optimization models is often a complex task that requires a collaborative effort between domain experts, who possess deep knowledge of the industry practices, and optimization experts, who are skilled in translating real-world problems into mathematical models and applying solution techniques Vineetha & Shiyas (2020). Such collaboration often demands significant time and expertise, which can pose a significant barrier to the widespread adoption of OR methods.

To address this challenge, there is an increasing demand for automated modeling tools that can bridge the gap between textual problem descriptions and mathematical models. Automating the modeling process can reduce reliance on experts, enhance accessibility for non-experts, and expedite decision-making processes Kushman et al. (2014) and Miyani et al. (2015). Such tools have the potential to democratize access to OR techniques, enabling a broader range of practitioners to leverage optimization in their operations.

Recent advancements in Large Language Models (LLMs), such as GPT-4 OpenAI et al. (2023) and Llama-3 Dubey et al. (2024), have demonstrated remarkable capabilities in understanding, reasoning, and planning Ouyang et al. (2022); Achiam et al. (2023); Radford et al. (2019); Song et al. (2024). These models have been successfully applied in various domains, including code generation Chen et al. (2021) and mathematical theorem proving Yang et al. (2023), showcasing their potential in tasks that require comprehension of complex language and logical structures. Consequently, there have been attempts to leverage LLMs for automatic optimization modeling, including LLM-based agent Xiao et al. (2023); AhmadiTeshnizi et al. (2024) and fine-tuning LLM for optimization modeling Tang et al. (2024). However, benchmarking the modeling ability of LLMs in OR remains a challenging endeavor.

One significant challenge is the absence of comprehensive benchmark datasets specifically designed for evaluating the optimization modeling capabilities of LLMs. Existing datasets often include both problem description and numerical data in the prompt Ramamonjison et al. (2022a); Xiao et al. (2023). This setting extremely limits the problem size, which is not reflective of practical modeling problems where data is typically large-scale and independent of the modeling process ApIO et al.. Moreover, these datasets lack comprehensiveness, often evaluating models over a single or limited level of complexity, and failing to capture the diverse range of problems encountered in real-world applications.

Another challenge lies in the evaluation process, which is frequently imprecise and time-consuming. Verification of the generated models often relies solely on solving the optimization problem and comparing the solutions AhmadiTeshnizi et al. (2024); Xiao et al. (2023), but many Mixed-Integer Linear Programming (MILP) problems are NP-hard and may not reach globally optimal solutions within a reasonable timeframe Chen et al. (2022b). This reliance on solvers for verification makes it difficult to accurately assess the correctness and quality of the generated models, as suboptimal solutions or solver timeouts can obscure the evaluation Han et al. (2023).

In this paper, we address these challenges by introducing a comprehensive benchmark dataset and an evaluation tool grounded in a principled and theoretically supported paradigm. Our contributions are threefold:

1. **Benchmark Dataset with Comprehensive Problem Features**: We develop a set of 816 word problems with a clear separation between model and data, closely mimicking real-world business problems. The dataset is constructed through a hierarchical reverse data evolution pipeline, allowing controlled generation of problem instances with varying complexity across 16 problem classes in LPs and MILPS, over 80 practical domains, and 2 levels of abstraction. Each problem is rigorously verified by OR experts to ensure accuracy and relevance. This approach enables a more practical and thorough assessment of LLMs' modeling capabilities across multiple aspects, including problem size, constraints, and objective functions.

2. **Benchmark Evaluation Tool with Principled and Theory-Supported Paradigm**: We propose a novel evaluation framework that formalizes the assessment of optimization modeling based on graph theory. By representing mathematical models as graph structures, we enable a structured and precise comparison between the generated models and ground truth models. We develop an automatic evaluation algorithm that efficiently computes similarity metrics between graphs, with proven computational efficiency and scalability. This method overcomes the limitations of solver-based verification by providing a direct evaluation of model correctness and completeness.

3. **Benchmarking Popular LLMs and Recent Agent-Based Approaches**: We conduct extensive experiments evaluating the performance of leading LLMs, including GPT-4o OpenAI et al. (2023), and other models such as o1 OpenAI (2024), Llama-3 Dubey et al. (2024), Claude Anthropic (2024a;b) on our benchmark dataset. Our results demonstrate that GPT-4o outperforms other LLMs, including o1-preview in optimization modeling tasks, highlighting its superior capability in understanding and formulating complex optimization problems. Interestingly, existing agent-based methods, which involve iterative reasoning or decomposition strategies, did not show significant improvements over direct prompting methods. We provide a comprehensive error analysis across various problem features, shedding light on the strengths and limitations of current LLMs in optimization modeling.

Our work aims to advance the field of automated optimization modeling by providing essential tools and datasets for rigorous evaluation and by illuminating the current capabilities and limitations of LLMs in this domain. We believe that our contributions will stimulate further research and development in automated decision-making systems, ultimately making OR techniques more accessible and practical for a wider audience.

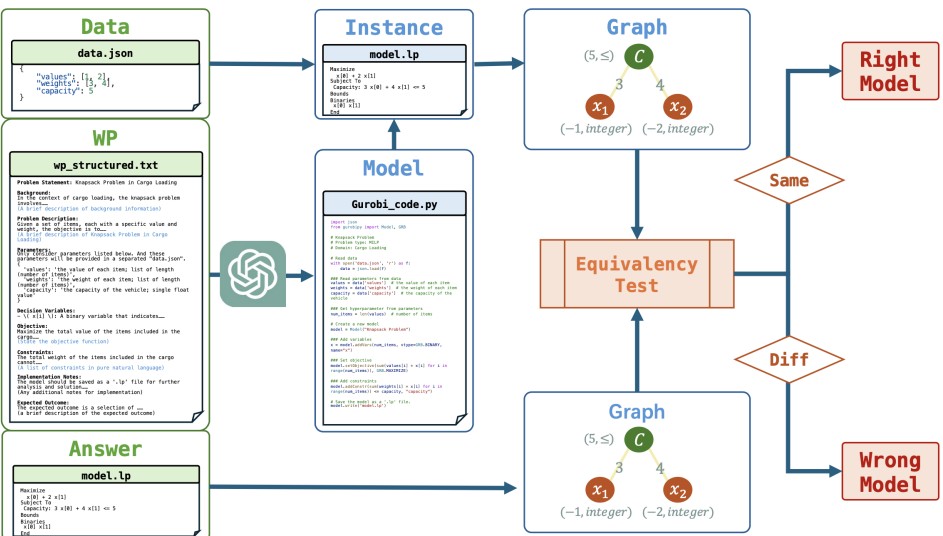

Figure 1: Benchmarking Pipeline

## 2 BACKGROUND AND RELATED WORK

### 2.1 BACKGROUND

Optimization problems are fundamentally characterized by an objective function that needs to be minimized, subject to a set of constraints involving decision variables and parameters. In this work, we focus on classical problems in linear programming (LP) and mixed-integer linear programming (MILP). In the following of this paper, we denote a MLLP/LP problem $\mathcal{P}$ by

$$\mathcal{P}: \min_{\mathbf{x} \in \mathbb{R}^p \times \{0,1\}^{n-p}} \mathbf{c}^T \mathbf{x}, \tag{1}$$
$$\text{s.t. } \mathbf{Ax} \circ \mathbf{b}, \mathbf{l} \leq \mathbf{x} \leq \mathbf{u},$$

where $\mathbf{A} \in \mathbb{R}^{m \times n}$, $\mathbf{c} \in \mathbb{R}^n$, $\mathbf{b} \in \mathbb{R}^m$, $\circ \in \{=, <, >, \leq, \geq\}^m$, $l_i, u_i \in \mathbb{R} \cup \{\infty, -\infty\}, \forall i = 1, \cdots, n$.

The modeling procedure in optimization typically involves a collaborative three-stage process between domain experts and operations research professionals. In the **pre-modeling stage**, the business problem is articulated in natural language, and all relevant data are collected, including numerical values for parameters and coefficients essential for model formulation. The **modeling stage** involves abstracting this problem description into a mathematical model, defining decision variables, formulating the objective function, and establishing the constraints that capture the problem's essence. Finally, in the **post-modeling stage**, the abstracted model is translated into solver-ready code, integrating the collected data to produce a fully specified problem realization ready for computational solving. This systematic approach ensures that complex real-world problems are accurately and efficiently translated into mathematical models for optimal decision-making.

### 2.2 RELATED WORK

**NLP for OR Modeling**   While substantial progress has been made in automatic modeling of general mathematical problems Bobrow (1964); Dellarosa (1986); Sundaram & Khemani (2015), there

has been limited focus on applying these techniques specifically to operations research. Prior to the rise of LLMs, the NL4Opt competition Ramamonjison et al. (2022b) explored the feasibility of learning-based natural language interfaces for optimization solvers. More recently, works leveraging LLMs, such as the Chain-of-Experts (CoE)Xiao et al. (2023) and OptiMUS AhmadiTeshnizi et al. (2024), introduced multi-agent cooperative systems to model and code complex OR problems automatically. Furthermore, Tang et al. (2024) fine-tuned open-source LLMs with approximately 7 billion parameters, achieving significant performance improvements over baseline models. These advancements underscore the immense potential of LLMs to enhance the efficiency and accuracy of optimization modeling tasks, paving the way for automated natural language interfaces for optimization solvers.

**Broader Research on AI for OR**  Beyond model formulation, significant progress has been made in the field of AI for Operations Research (AI for OR), particularly in parameter generation and model optimization Rajgopal (2004). In parameter generation, AI techniques have been employed for better simulation of key parameters of optimization problems Elmachtoub & Grigas (2022); Maragno et al. (2023); Bergman et al. (2022). Similarly, we leverage LLMs to generate necessary problem data through a program of thoughts Chen et al. (2022a). On the optimization side, numerous studies have focused on leveraging AI models in automatic algorithm configuration Ansótegui et al. (2009); Lindauer et al. (2022); Anastacio & Hoos (2020) , optimization algorithm selection Wang et al. (2019); Chi et al. (2022), and heuristic algorithm design Zeng et al. (2022); Talbi (2009); Romera-Paredes et al. (2024). Specifically, a line of research has modeled MILP/LP problems as bipartite graphs and applied Graph Neural Networks (GNNs) to make decisions at various stages of their solution processesGasse et al. (2019); Zhou et al. (2020). These GNN-based methods have demonstrated efficacy in tasks such as variable selection and node branching, leading to significant improvements in solver performance. Inspired by this, we model optimization problems as bipartite graphs and formalize the evaluation paradigm based on the classical Weisfeiler-Lehman graph isomorphism test (WL-test) algorithm Leman & Weisfeiler (1968).

**Benchmarking LLM on complex tasks**  With the emergence of LLMs, there is an increasing need for benchmarks to understand their capability boundaries Liu et al. (2024); Zhou et al. (2024); Sawada et al. (2023). Several optimization modeling benchmarks have been proposed to evaluate LLMs. The Linear Programming Word Problem (LPWP) dataset Ramamonjison et al. (2022a)includes multiple domains and comprises up to 1,001 LP problems. However, it primarily consists of elementary-level LP problems, limiting its effectiveness in assessing advanced modeling capabilities. The ComplexOR dataset Xiao et al. (2023) was designed to feature more intricate OR problems, but its limited size and inclusion of numerical data within problems constrain the level of complexity it can represent. The NLP4LP dataset AhmadiTeshnizi et al. (2024) attempts to separate data from model descriptions to provide a clearer evaluation of modeling skills, yet it remains small, with problems that are overly structured and explicitly described. Datasets like IndustryOR Tang et al. (2024), MAMO Huang et al. (2024), and E-OPT Yang et al. (2024) strive to cover a broader range of OR problems through data synthesis and augmentation. Nevertheless, they rely on solver solution-based evaluations, which fail to directly assess the modeling failure of LLMs and are limited by solvers' performance. In comparison to existing benchmarks, our work aims to provide a more comprehensive dataset and to formalize the evaluation of modeling, enabling a more precise assessment of LLM capabilities in optimization modeling.

## 3 OPTIBENCH

To evaluate the potential of LLMs as interfaces for optimization solvers, particularly in the modeling and post-modeling stages we introduce OptiBench, a comprehensive benchmark designed to assess LLM capabilities in optimization modeling rigorously. OptiBench leverages LLMs to simulate real-world OR problems with delicate prompt engineering and expert verification. It evaluates LLMs' abilities in language comprehension, model formulation, and domain-specific coding tasks specific to optimization practices. Inspired by INFORMS modeling competition AIMMS (2024), we adopted a model-data separated format of the word problems.

The OptiBench encompasses multi-dimensional complexity through a hierarchical reverse data evolution, including optimization problem **type**, **classes**, **domains**, **variants**, and **level of abstraction**. Our dataset comprises 816 optimization modeling word problems of two fundamental types

of optimization problems—LPs and MILPs. The dataset spans over 16 classical optimization problem classes, including but not limited to cutting stock(LP), network flow(LP), bin-packing problems(MILP), and set covering problem(MILP), detailed classes see Table 2. To examine LLMs' ability for practical relevance, the dataset spans about 80 class-specific domains, such as telecommunication, transportation, and supply chain management, reflecting real-world scenarios across diverse industries. With approximately 480 optimization models, each represented in 2 levels of description abstraction, OptiBench offers a wide array of challenges requiring domain knowledge and a nuanced understanding of optimization.

## 3.1 MODEL-DATA SEPARATION

We adopt a model-data-separated format to mirror real-world optimization modeling tasks in our word problems. This reflects standard practices in industrial settings, where problem formulation and data collection are often sequential and handled by different teams.

Each word problem (WP) in the dataset consists of

- **Problem Description ('wp.txt')**: A detailed description of the optimization scenario, including objectives and constraints, without embedding specific numerical data, and a specification of the nature and structure of the required data, guiding the LLM on what information is needed without providing actual values.

- **Data File ('data.json')**: A structured file containing all the numerical data necessary to model and solve the problem. This separation ensures that LLMs formulate the problem based on the description before applying the data.

- **Reference Model ('model.lp')**: A reference answer of the modeling problem in '.lp' format, structured for readability of specific problem instances and easy exporting and importing using optimization solvers.

An illustrative example is provided in Figure 2, demonstrating how the problem description and data file complement each other. A example of our word problem is listed in **??**By decoupling problem complexity from data size, we can create conceptually challenging problems without being prohibitive by LLM's context window size.

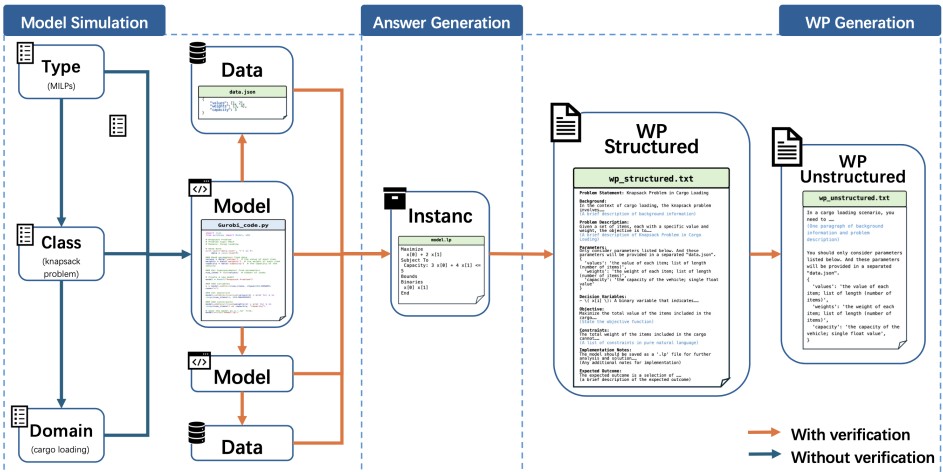

Figure 2: Data Construction Pipeline

## 3.2 MULTI-DIMENSIONAL COMPLEXITY

To construct the multidimensional complexity efficiently, we employ a hierarchical reverse data evolution pipeline encompassing three key stages: optimization model stimulation, reference answer generation, and word problem generation.

**Optimization Models Stimulation**    We leverage GPT-4o to propose optimization models, systematically evolving the context through problem types, classes, domains, and variants. Specifically, GPT-4o first identifies classical problem classes within LP and MILP. For each problem class, GPT-4o subsequently determines common application domains, ensuring relevance and applicability. Within each domain-specific problem class, GPT-4o first proposes the most canonical optimization model and then augments it by varying optimization components to introduce diversity and complexity.

**Reference Answers Generation**    The simulated optimization model is represented as Gurobi code that reads from an associated data file 'data.json' and outputs the corresponding realization as 'model.lp'. This approach facilitates the evaluation process by providing a standardized reference answer. To create the necessary 'data.json' files, we utilize GPT-4o to generate Python programs tailored to each optimization model. This method allows for scalable problem size adjustments by simply modifying the dimensionality instructions within the prompts, thereby easing the expansion of specific optimization models.

**Word Problems Generation**    Finally, we reversely generate word problems from the stimulated optimization models. Drawing insights from the INFORMS AIMMS-MOPTA AIMMS (2024) Optimization Modeling Competition, we meticulously crafted a standardized word problem structure, as illustrated in Figure 5. Using GPT-4o, we first translate the solver code into a detailed word problem adhering to this standardized structure. Subsequently, we refine and summarize the generated content to produce more concise and unstructured problem statements. This comprehensive approach ensures that the generated word problems are both accurate representations of the underlying optimization models and suitable for benchmarking in a more complex manner.

### 3.3 QUALITY CONTROL

We implement controlled generation with meticulous verification to ensure data quality in our benchmark.

**Controlled Generation**    Similar to crafting standard word problem structures, we construct a code skeleton specifically designed to stimulate optimization models. This simplifies the free-generation task into a more manageable code completion task, which allows us to regulate the LLM's output tightly and thereby partially automate data evolution and verification pipeline. This controlled framework maintains consistency across generated data and ensures that the models adhere to the necessary structural requirements, as exemplified in Figure 8.

**Verification**    We rigorously assess the validity of each model-answer pair. We execute the code representations during the model simulation and answer generation phase; only those that run without errors are considered valid. Additionally, in the word problem generation stage, we employ an LLM-based verifier to ensure the optimization components in the code and their corresponding elements in the word problems are precisely matched. Experts in operations research further validated this and removed problematic instances from our dataset.

Through our controlled generation and thorough verification, we maintain high data quality and accuracy standards, ensuring the reliability and robustness of our benchmark for evaluating the capabilities of large language models in operations research modeling.

## 4 EVALUATION PARADIGM

### 4.1 EVALUATION PRINCIPAL

To evaluate if the LLM gives the correct answer for optimization modeling, we combine their generation with problem data to form a test modeling instance and compare the test instance with the corresponding standard instance in our benchmark.

In general, it is challenging to check whether two optimization problem instances are "equal". For example, variables can be named in different notations or presented in different orders, resulting in a set of not exactly the same but equivalent modeling. To fill this gap, we propose a new evaluation paradigm, which identifies the correctness of optimization modeling by detecting whether the inherent structure of the test instance is equivalent to that of the standard instance.

We first establish the correctness for MILP and LP model formulation based on the following three principles: (1) Variables should reflect real-world entities. (2) Objectives should clearly align with their descriptions, and (3) Constraints should represent real-world limitations without redundancy.

A model following the above principles should be regarded as correct. We further introduce the notion of **equivalence** between two model instances. Specifically, the equivalence allows model instances to change the notations of variables and rearrange their variables/constraints without losing essential information. We denote that two instances $\mathcal{P}_1$ and $\mathcal{P}_2$ are model-equivalent by $\mathcal{P}_1 \sim \mathcal{P}_2$; a formal definition is provided in Appendix A.2.

Our concept of model-wise equivalence aligns with the isomorphism in graphs, allowing nodes to be re-indexed or rearranged without changing the graph structure. This motivates us to incorporate tools in graph theory to evaluate model equivalence. Following existing work in the field of learning to optimize Gasse et al. (2019); Chen et al. (2022b), we represent an LP/MILP model realization with a bipartite graph (Figure 1). We proved that detecting the model equivalence can be reduced to testing graph isomorphic; See Appendix A.4 for a formal demonstration.

## 4.2 EVALUATION METHOD

Based on our equivalence metric, we evaluate modeling result in two steps:

**Create test and standard graphs** As in Figure 1, we represent MILP/LP instances as bipartite graphs. In such graphs, nodes can be divided into two groups– variable nodes and constraint nodes. All nodes are equipped with the necessary features. Each constraint node connects with all associated variable nodes. We follow the formal notation from Chen et al. (2022b); the detailed definition is presented in Appendix A.3.

**Isomorphism testing** Graph isomorphism testing is a challenging problem, with no known polynomial-time algorithm to date Garey & Johnson (1979); Babai (2016). Except for some corner cases Cai et al. (1992), the Weisfeiler-Lehman (WL) test of graph isomorphism Leman & Weisfeiler (1968) is an effective and computationally efficient method for distinguishing a wide range of graphs. Typically, one may determine that two graphs are non-isomorphic if the WL test algorithm produces different coloring distributions. However, if the WL test yields the same distribution for two graphs, it does not guarantee that the graphs are isomorphic. To prevent misjudgment, we propose a modified isomorphism testing algorithm for equivalence detection; see Algorithm 1.

---

**Algorithm 1** Modeling Equivalence Detection

**Require:** Two graph instances $(G_k, H_k) \in \mathcal{G}_{m,n}^k \times \mathcal{H}_m^V \times \mathcal{H}_n^W$ and adjacency matrix $\mathbf{A}_k, k = 1, 2$; iterate limit $L > 0$.
1: Color nodes in two graphs using WL-test Algorithm for MILP/LP, get two coloring multi-sets $\mathcal{C}_k = \left\{ \{\{C_i^{k,V}\}\}_{i=0}^m, \{\{C_j^{k,W}\}\}_{j=0}^n \right\}, k = 1, 2$ for coloring $\mathcal{G}_1$ and $\mathcal{G}_2$.
2: Derive set of unique elements in $\mathcal{C}_k$, denote as set $\mathbb{A}_k, \forall k = 1, 2$.
3: **if** $\mathcal{C}_1 \neq \mathcal{C}_2$ **then**
4:     **return** Not same
5: **else if** $len(\mathbb{A}_1) = len(\mathcal{C}_1)$ & $len(\mathbb{A}_2) = len(\mathcal{C}_2)$ **then**     ▷ Check sufficient condition 1
6:     **return** Same
7: **else if** $len(\mathbb{A}_1) \neq len(\mathcal{C}_1)$ **then**
8:     **if** $\mathcal{G}_1$ is symmetric decomposable [1] **then**     ▷ Check sufficient condition 2
9:         **return** Same
10:     **else**
11:         **return** Not Same
12:     **end if**
13: **end if**

---

This algorithm involves running a WL-test for MILP in the first step, we use the same implementation as Chen et al. (2022b); See Algorithm 2. After getting the coloring distribution from the WL test, the algorithm checks whether the two instances satisfy any sufficient conditions, such that these graphs can be discriminated directly through the coloring distribution; see Algorithm 3.

We proved that our modeling equivalence detection algorithm can test isomorphism for all graph instances in our benchmark. In addition, in our benchmark, the time complexity to distinguish tested problem instances from the standard instances with $m$ variables and $n$ nodes is at most $\mathcal{O}(mn + n\log n)$, this is far better than complexity for exhausted isomorphism testing; detailed complexity analysis can be found in Appendix A.7.

### 4.3 Theoretical Guarantee

Though the WL-test is widely used for isomorphism testing, it may fail to distinguish non-isomorphic graphs in certain exceptional cases; counter-examples are presented in Appendix A.11 Chen et al. (2022b).

In previous work, Chen et al. (2022b) characterized one sufficient condition of problem instances, say unfoldable, that can be accurately distinguished by WL-Test. Yet it is too strict for many MILP problems. For example, graphs for bin-packing instances are typically not unfoldable; see Appendix A.11 for an example. We extend the sufficient conditions to cover more cases, which benefits the problems encountered in our benchmark.

To clarify the sufficient conditions for WL test in our evaluation paradigm, we define a class of **WL-determinable** and **decomposable symmetric** problem instances.

**Definition 4.1 (WL-Determinable Instance)** *We say a model instance $\mathcal{P}$ is **WL-determinable** if WL test outputs distinct colors for different nodes in its graph representation.*

This definition aligns with the definition of unfoldable graphs.

**Definition 4.2 (Decomposable Symmetric Instance)** *We say a modeling instance $\mathcal{P}$ is decomposable symmetric if, after WL test coloring on its representation graph, the following conditions hold:*

1. *Excluding nodes that have distinct colors from all other nodes, the remaining nodes can be divided into groups, denoted by $I_1, I_2, \cdots, I_k$, each containing at least two nodes. All nodes in the same group share the same color.*

2. *For any pair of groups $I_i$, $I_j$, either $I_i$ and $I_j$ are disconnected, or the nodes in $I_i$ and $I_j$ form a perfect matching. Specifically, a perfect matching means that every node in $I_i$ is connected to exactly one node in $I_j$, and every node in $I_j$ is connected to exactly one node in $I_i$.*

One example of a decomposable symmetric instance can be found in Figure 10.

In the following theorem, we showed that if the standard instance satisfies either of the two sufficient conditions: being WL-determinable or decomposable symmetric, then Algorithm 1 can be reliably used for detecting whether a test instance is model-equivalent to the standard instance. Rigorous proof can be found in Appendix A.8.

**Theorem 4.1** *Denote Algorithm 1 by $\mathcal{A}(\mathcal{G}_{test}, \mathcal{G}_{standard})$. Suppose $\mathcal{P}_{standard}$ is WL-determinable or decomposable symmetric, then $\forall \mathcal{P}_{test}$, we have $\mathcal{A}(\mathcal{G}_{test}, \mathcal{G}_{standard}) == True \iff \mathcal{P}_{test} \sim \mathcal{P}_{standard}$.*

**Generality of WL-determinable and Symmetric Decomposable Instances** Many operations research models, such as assignment problems and traveling salesman problems, are almost surely to get a WL-determinable instance by random sampling problem data, we provide a theorem in Appendix A.9 to characterize this property. Our algorithm can determine whether a problem is WL-determinable or Symmetric Decomposable by definition. Empirically, although we did not intentionally select models and problem data for our benchmark, we found that almost all instances in our benchmarks are either WL-determinable or symmetric-decomposable.

## 5 Experiment and Analysis

**Experiment Setting** To assess the capabilities of LLMs in optimization modeling, we conducted a comprehensive evaluation using the OptiBench benchmark. Our evaluation focused on top-

performing LLMs via direct prompting, including closed-sourced models such as GPT-4o OpenAI et al. (2023), o1-preview, o1-mini OpenAI (2024), Claude-3.5-sonnet Anthropic (2024a), Claude-3-opus Anthropic (2024b), and the open-sourced LLM Llama-3-70b-instruct Dubey et al. (2024). We also deployed the Chain of Expert modeling agent Xiao et al. (2023). Each LLM was tested across all 816 OptiBench questions to ensure a thorough and consistent assessment of their optimization modeling abilities. The main evaluation result is listed in Table 1

Table 1: Evaluation Results on OptiBench. The "Overall" modeling accuracy is the accuracy weighted by question count. The SOTA in each category is marked in red.

| LLMs | Modeling Accuracy | | | | |
| | LP | | MILP | | Overall |
| | Structured | Unstructured | Structured | Unstructured | |
| Direct Prompting | | | | | |
| gpt-4o | 56.87 | 42.18 | 56.85 | 41.62 | 49.39 |
| o1-preview | 47.87 | 32.70 | 43.65 | 30.46 | 38.73 |
| o1-mini | 45.97 | 35.55 | 43.15 | 32.99 | 39.46 |
| claude-3-5-sonnet | 45.97 | 33.18 | 51.78 | 39.59 | 42.52 |
| claude-3-opus | 52.61 | 39.34 | 51.78 | 34.52 | 44.61 |
| llama3-70b-instruct | 42.65 | 24.17 | 39.09 | 29.44 | 33.82 |
| LLM-based Agent | | | | | |
| Chain-of-Experts | 48.82 | 37.91 | 51.27 | 30.47 | 43.50 |

**Comparing Performance Across Different LLMs and Prompting Methods**    Among the evaluated models, GPT-4o achieved the highest overall performance, securing an accuracy rate of 45.38% across all problem categories. Surprisingly, both the o1-preview and o1-mini models underperformed GPT-4o. Claude-3.5-sonnet outperformed both o1-preview and o1-mini in MILPs, while Claude-3-opus surpassed o1-preview and o1-mini across all tested settings.

Furthermore, the application of the Chain of Expert agents, intended to enhance problem-solving through multi-agent collaboration and extensive reasoning paths, inadvertently reduced the performance of GPT-4o to 39.98%. The intended multi-step reasoning in both o1 and LLM-based agents may have introduced inconsistencies in the generated code, which decreased the code pass rate. Additionally, the accumulation of hallucinations—incorrect or fabricated information—further exacerbated performance degradation, ultimately lowering the overall accuracy.

Note that we also explored the performance of the OptiMUS model AhmadiTeshnizi et al. (2024). Initially, OptiMUS showed extremely bad performance due to several reasons. First, OptiMUS requires extracting optimization entities during the initial modeling phase. However, the extraction accuracy on OptiBench is below 50%, which stops the agent from moving toward subsequent modeling steps.(We monitor the sanity check and OptiMUS's early interruption primarily due to parameter names or dimensions mismatches. To mitigate this issue, we designed an improved extraction agent tailored for OptiMUS. However, despite this enhancement, the overall code pass rate remained below 10%, leading to an overall accuracy below this threshold.

**Comparing Performance Across Different Dimensions of Complexity**    Our analysis revealed significant variations in LLM performance based on the complexity dimensions of the OR problems. Specifically, MILPs were consistently more challenging for the LLMs compared to LPs. This increased difficulty is likely due to the combinatorial nature and higher computational complexity inherent in MILPs formulations. Furthermore, unstructured problems posed a more significant challenge than structured ones, indicating that LLMs struggle more with tasks that lack clear formatting or predefined frameworks. Both Llama-3-70b-instruct and Claude-3.5-sonnet demonstrated comparable performance levels on the unstructured versions of LP and MILP tasks.

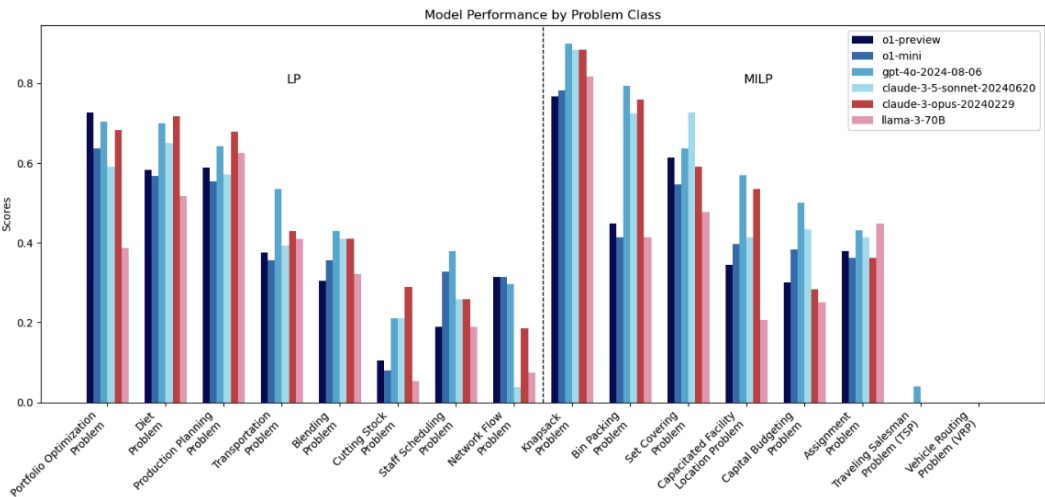

Figure 3: Performance Across Different Classes

**Comparing Performance Across Different Problem Classes**  Beyond the primary performance metrics, we also examined the modeling accuracy across various classes of OR problems to identify potential biases in LLM knowledge bases. The results indicated a pronounced bias, with relatively high accuracy observed in solving knapsack problems (MILP), assignment problems (MILP), and diet problems (LP). These areas likely benefit from the simplicity of problem model and more extensive representation in training data. Conversely, the models exhibited relatively low accuracy in addressing cutting stock problems (MILP) and near-zero accuracy for traveling salesman problem and vehicle routing problem. These findings underscore significant gaps in LLMs' capabilities, particularly in handling specialized and highly complex OR problems. The observed biases suggest that while LLMs are proficient in certain well-represented problem classes, their effectiveness diminishes in less common or more intricate problem spaces, highlighting areas for future research and training improvement.

## 6 CONCLUSION

In this work, we introduced **OptiBench**, a novel benchmark to evaluate the ability of LLMs in optimization modeling tasks. OptiBench uniquely incorporates multi-dimensional complexity in a model-data-separated manner, allowing a more structured and flexible evaluation process. To facilitate a comprehensive assessment of LLMs' optimization capabilities, we formalized an evaluation paradigm based on equivalence detection, ensuring accurate and meaningful comparisons between models. We also theoretically proved the efficiency of our proposed method. By benchmarking over OptiBench, GPT-4o demonstrated superior performance in the direct prompting setting, outperforming all other LLMs and agents. In contrast, the latest model, o1-preview, and the existing modeling agent surprisingly underperformed compared to GPT-4o. This underperformance might be attributed to the snowball effect of hallucination, especially prevalent during longer reasoning paths when tackling complex tasks. Our results suggest that while current LLMs possess a foundational capability in optimization modeling, there remains significant room for improvement. We plan to develop a specialized modeling agent to address these gaps, incorporating a curated reasoning skeleton tailored specifically for optimization and operational research. In addition, we intend to extend our hierarchical reverse data evolution method to create fine-tuning datasets for optimization tasks and broader logical reasoning tasks. Through these efforts, we aim to push the boundaries of LLMs' operational research and optimization modeling capabilities, ultimately fostering advancements in AI research and practical applications.

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
