# A  APPENDIX

## A.1  DATASET

Table 2: Optimization problem types and classes including in our OptiBench.

| Problem Types | Problem Classes |
| --- | --- |
| LPs | Diet Problem |
| | Transportation Problem |
| | Blending Problem |
| | Production Planning Problem |
| | Network Flow Problem |
| | Portfolio Optimization Problem |
| | Cutting Stock Problem |
| | Staff Scheduling Problem' |
| MILPs | Knapsack Problem |
| | Traveling Salesman Problem (TSP) |
| | Vehicle Routing Problem (VRP) |
| | Bin Packing Problem |
| | Set Covering Problem |
| | Capacitated Facility Location Problem |
| | Capital Budgeting Problem |
| | Assignment Problem |

## A.2  MODEL EQUIVALENCE CLASS

**Definition A.1 (Model Equivalence)**  *We say $\mathcal{C}(\mathcal{P})$ is a **model equivalence class** of the MILP/LP problem instance $\mathcal{P}$ if $\forall \hat{\mathcal{P}} \in \mathcal{C}(\mathcal{P}), \exists$ permutation matrices $P_1, P_2$ which shuffles the index of a vector or column index of a matrix s.t. $\hat{\mathcal{P}}$ can be written in the following form:*

$$\min_{x} \hat{c}^T x,$$
$$s.t. \ \hat{A}x \hat{\circ} \hat{b}, \hat{l} \le x \le \hat{u}$$

*where $\hat{b} = P_2 b, \hat{C} = P_1 C, \hat{A} = P_2 A P_1, \hat{\circ} = P_2 \circ, \hat{l} = P_1 l, u = P_1 u$.*

*$\forall \mathcal{P}_2 \in \mathcal{C}(\mathcal{P}_1)$, we say $\mathcal{P}_2$ is **model-equivalent** to $\mathcal{P}_1$, denote as $\mathcal{P}_1 \sim \mathcal{P}_2$.*

## A.3  WEIGHTED BIPARTITE GRAPH FOR REPRESENTING MILP/LP

A weighted bipartite graph for a MILP/LP instance is denoted by $\mathbf{G} = (\mathbf{V} \cup \mathbf{W}, \mathbf{E})$, with vertex set $\mathbf{V} \cup \mathbf{W}$ divided into 2 groups $\mathbf{V} = \{\mathbf{v}_1, \cdots, \mathbf{v}_m\}$ for constraints, and $\mathbf{W} = \{\mathbf{w}_1, \cdots, \mathbf{w}_n\}$ for variables, $\mathbf{E}$ consisting of $E_{ij} = E(v_i, w_j), \forall i = 1, \cdots, m, j = 1, \cdots, n$. To fully represent all information in a MILP/LP instance, we associate each vertex with features:

- The constraint vertex $\mathbf{v}_i \in \mathbf{V}$ is equipped with a feature vector $\mathbf{H}^V$ with elements $\mathbf{h}_i^V = (b_i, o_i) \in \mathcal{H}^V = \mathbb{R} \times \{\le, \ge, =, <, >\}$
- The variable vertex $\mathbf{w}_j \in \mathbf{W}$ is equipped with a feature vector $\mathbf{H}^W$ with elements $\mathbf{h}_j^W = (c_j, l_j, u_j, \tau_j) \in \mathcal{H}^W = \mathbb{R} \times \{\mathbb{R} \cup -\infty\} \times \{\mathbb{R} \cup \infty\} \times \{0, 1\}$. $\tau_j = 1$ if $j \in \mathbb{I}$ and $\tau_j = 0$ otherwise.

The edge $E_{ij} \in \mathbb{R}$ connects $\mathbf{v}_i \in \mathbf{V}$ and $\mathbf{w}_j \in \mathbf{W}$, $E_{ij} = \mathbf{A}_{ij}$. There is no edge connecting vertices in the same vertex group.

```python
import json
from gurobipy import Model, GRB

# {problem_class} — {problem_name}
# Problem type: {problem_type}
# Domain: {domain}
# Property: (fill in this comment by briefly describing the
variant of the problem)

# Read data
with open('data.json', 'r') as f:
data = json.load(f)

### (fill in this section) Read parameters from data (assign
domain specific parameter name)

### (fill in this section) Get hyperparameter from parameters
(assign domain specific parameter name)

# Create a new model
model = Model("{problem_class}")

### (fill in this section) Add variables of the classic
{problem_name} (assign domain specific name)

### (fill in this section) Set objective of the {problem_name}
(assign domain specific name)

### (fill in this section) Add constraints of the
{problem_name} (assign domain specific name)

# Save the model as a '.lp' file.
model.write('model.lp')
```

Figure 4: Code skeleton for optimization model simulation.

## A.4 Connection between Model Equivalence and Graph Isomorphism

To test whether 2 modeling instances were permutation equivalent, we can equivalently conduct isomorphism testing between their corresponding weighted bipartite graphs. Lemma A.1 establishes an equivalence between assessing modeling appropriateness and graph isomorphism testing.

**Definition A.2 (Graph Isomorphism)** *Consider 2 graphs* $\mathcal{G}_1 = (\mathbf{G}_1, \mathbf{H}_1^V \times \mathbf{H}_1^W)$ *and* $\mathcal{G}_2 = (\mathbf{G}_2, \mathbf{H}_2^V \times \mathbf{H}_2^W)$ *with* $\mathbf{G}_i = (\mathbf{V}^i \cup \mathbf{W}^i, \mathbf{E}^i)|_{1 \leq i \leq 2}$. *We say* $\mathcal{G}_1$ *and* $\mathcal{G}_1$ *are* **isomorphic** *if there exists permutation matrix* $\mathbf{P}_1, \mathbf{P}_2$ *such that:* $\mathbf{P}_1 \mathbf{E}^1 P_2^T = \mathbf{E}^2, \mathbf{P}_1 \mathbf{H}_1^W = \mathbf{H}_2^W, \mathbf{P}_2 \mathbf{H}_1^V = \mathbf{H}_2^V$. *If 2 graphs* $\mathcal{G}_1$ *and* $\mathcal{G}_1$ *are isomorphic, denote* $\mathcal{G}_1 \overset{g}{\sim} \mathcal{G}_2$.

**Lemma A.1** $\forall$ *MILP/LP instances* $\mathcal{P}_1, \mathcal{P}_2$ *with corresponding bipartite graph* $\mathcal{G}_1, \mathcal{G}_1$, *we have*

$$\mathcal{P}_1 \sim \mathcal{P}_2 \Longleftrightarrow \mathcal{G}_1 \overset{g}{\sim} \mathcal{G}_2.$$

```
**Problem Statement: {problem_name} in
{domain}**

**Background:**
(A brief description of background
information)

**Problem Description:**
(A brief description of {problem_name} in
{domain})

**Parameters:**
Only consider parameters listed below. And
these parameters will be provided in a
separated "data.json".
{parameter_skeleton}

**Decision Variables:**
(A list of decision variables and their
description)

**Objective:**
(State the objective function)

**Constraints:**
(A list of constraints in pure natural
language)

**Implementation Notes:**
(Any additional notes for implementation)

**Expected Outcome:**
(a brief description of the expected outcome)
```

Figure 5: Standard structure for word problem crafted from INFORMS AIMMS-MOPTA Optimization Modeling Competition.

A.5  PROOF OF LEMMA A.1:

We prove this lemma by proving 2 claims:

**Claim 1:**  $\mathcal{G}_1 \sim \mathcal{G}_2 \implies \mathcal{P}_1 \sim \mathcal{P}_2$.

Suppose $\mathcal{G}_1 \sim \mathcal{G}_2$. For bipartite graphs $\mathcal{G}_1$ and $\mathcal{G}_2$, nodes $v_i$ would only connect with some node $w_j$ if the $j$-th constraint involves decision variable $x_i$. Therefore the adjacency matrix of $\mathcal{G}_k$ would be in the form $\mathbf{A}_{\mathbf{adj}}^{(\mathbf{k})} = \begin{bmatrix} 0 & \mathbf{A}_k^T \\ \mathbf{A}_k & 0 \end{bmatrix}, \forall k = 1, 2$. Now, by the assumption that $\mathcal{G}_1 \sim \mathcal{G}_2$, $\exists$ permutation

```
**Problem Statement: Knapsack Problem in Cargo Loading**

**Background:**
In the context of cargo loading, the knapsack problem involves selecting a subset of items
to include in a cargo such that the total value of the selected items is maximized, while
ensuring that the total weight of the selected items does not exceed the vehicle's capacity.
This problem is a classical example of a combinatorial optimization problem and is widely
studied in operations research.

**Problem Description:**
Given a set of items, each with a specific value and weight, the objective is to determine
which items to include in the cargo to maximize the total value without exceeding the
vehicle's weight capacity. The decision to include an item in the cargo is binary (either
the item is included or it is not).

**Parameters:**
Only consider parameters listed below. And these parameters will be provided in a separated
"data.json".
{
    'values': 'the value of each item; list of length (number of items)',
    'weights': 'the weight of each item; list of length (number of items)',
    'capacity': 'the capacity of the vehicle; single float value',
}

**Decision Variables:**
- \( x[i] \): A binary variable that indicates whether item \( i \) is included in the cargo
(1) or not (0).

**Objective:**
Maximize the total value of the items included in the cargo. This is achieved by summing the
product of the value of each item and its corresponding binary decision variable.

**Constraints:**
The total weight of the items included in the cargo cannot exceed the vehicle's capacity.
This is ensured by summing the product of the weight of each item and its corresponding
binary decision variable and ensuring that this sum does not exceed the given capacity.

**Implementation Notes:**
- The problem is formulated as a Mixed-Integer Linear Programming (MILP) problem.
- The decision variables are binary, indicating the inclusion or exclusion of each item.
- The model should be saved as a '.lp' file for further analysis and solution.

**Expected Outcome:**
The expected outcome is a selection of items that maximizes the total value while ensuring
that the total weight does not exceed the vehicle's capacity. The solution will provide the
optimal set of items to include in the cargo.
```

Figure 6: Example for word problem on cargo loading.

matrix $\mathbf{P}$ such that

$$\mathbf{P} = \begin{bmatrix} \mathbf{P}_V & 0 \\ 0 & \mathbf{P}_W \end{bmatrix},$$

$$\mathbf{P}\mathbf{A}_{adj}^{(1)}\mathbf{P} = \mathbf{A}_{adj}^{(2)},$$

$$\mathbf{P}_V^T\mathbf{H}_1^V = \mathbf{P}^T\mathbf{H}_2^V,$$

$$\mathbf{P}_W^T\mathbf{H}_1^W = \mathbf{P}_W^T\mathbf{H}_2^W.$$

Therefore, we have

$$\mathbf{A}_{adj}^{(2)} = \begin{bmatrix} 0 & \mathbf{P}_V\mathbf{A}_1^T \\ \mathbf{P}_W\mathbf{A}_1 & 0 \end{bmatrix} \text{ and } \mathbf{H}_2 = \begin{bmatrix} \mathbf{P}_V\mathbf{H}_1^V \\ \mathbf{P}_W\mathbf{H}_1^W \end{bmatrix}.$$

```
In a cargo loading scenario, you need to choose a subset of items, each with a given value
and weight, to maximize total value without surpassing the vehicle's weight capacity. The
decision to include an item is binary. You'll be given a list of item values, weights, and
the vehicle's capacity. Your task is to determine which items to include to achieve the
highest total value while staying within the weight limit.

You should only consider parameters listed below. And these parameters will be provided in a
separated "data.json".
{
  'values': 'the value of each item; list of length (number of items)',
  'weights': 'the weight of each item; list of length (number of items)',
  'capacity': 'the capacity of the vehicle; single float value',
}
```

Figure 7: Example for concise version word problem on cargo loading.

```python
import json
from gurobipy import Model, GRB

# {problem_class} — {problem_name}
# Problem type: {problem_type}
# Domain: {domain}
# Property: (fill in this comment by briefly describing the
variant of the problem)

# Read data
with open('data.json', 'r') as f:
data = json.load(f)

### (fill in this section) Read parameters from data (assign
domain specific parameter name)

### (fill in this section) Get hyperparameter from parameters
(assign domain specific parameter name)

# Create a new model
model = Model("{problem_class}")

### (fill in this section) Add variables of the classic
{problem_name} (assign domain specific name)

### (fill in this section) Set objective of the {problem_name}
(assign domain specific name)

### (fill in this section) Add constraints of the
{problem_name} (assign domain specific name)

# Save the model as a '.lp' file.
model.write('model.lp')
```

Figure 8: Code skeleton for optimization model simulation.

We may reformulate the MILP/LP instance $\mathcal{P}_2$ as follows:

$$\mathcal{P}_2: \quad \min_{\mathbf{x} \in \mathbb{R}^p \times \{0,1\}^{n-p}} \mathbf{c}^T \mathbf{P}_V \mathbf{x},$$

$$\text{s.t. } \mathbf{P}_W \mathbf{A} \mathbf{P}_V \mathbf{x} \circ \mathbf{P}_W \mathbf{b}, \mathbf{l} \leq \mathbf{P}_V \mathbf{x} \leq \mathbf{u},$$

By the definition of permutation equivalent, we say $\mathcal{P}_2 \sim \mathcal{P}_1$.

**Claim 2:** $\mathcal{P}_1 \sim \mathcal{P}_2 \implies \mathcal{G}_1 \sim \mathcal{G}_2$.

Suppose $\mathcal{P}_1 \sim \mathcal{P}_2$. By the definition of permutation equivalent class, $\exists$ permutation matrix $\mathbf{P}_1$ and $\mathbf{P}_2$ such that

$$\mathbf{A}_2 = \mathbf{P}_2\mathbf{A}_1\mathbf{P}_1$$
$$\mathbf{b}_2 = \mathbf{P}_2\mathbf{b}_1,$$
$$\mathbf{C}_2 = \mathbf{P}_1^T\mathbf{C}_1,$$
$$\mathbf{P}_2\circ_1 = \circ_2,$$

Therefore, the corresponding adjacent matrix in the bipartite graph of $\mathcal{P}_2$ is

$$\mathbf{A}_{adj}^{(2)} = \begin{bmatrix} 0 & \mathbf{A}_2^T \\ \mathbf{A}_2 & 0 \end{bmatrix}$$
$$= \begin{bmatrix} 0 & \mathbf{P}_1^T\mathbf{A}_1^T\mathbf{P}_2^T \\ \mathbf{P}_2\mathbf{A}_1\mathbf{P}_2 & 0 \end{bmatrix}$$
$$= \begin{bmatrix} \mathbf{P}_1^T & 0 \\ 0 & \mathbf{P}_2 \end{bmatrix} \begin{bmatrix} 0 & \mathbf{A}_1^T \\ \mathbf{A}_1 & 0 \end{bmatrix} \begin{bmatrix} \mathbf{P}_1 & 0 \\ 0 & \mathbf{P}_2^T \end{bmatrix}$$
$$= \hat{\mathbf{P}}^T\mathbf{A}_{adj}^{(1)}\hat{\mathbf{P}}$$

In addition, we have $\mathbf{b}_2 = \mathbf{P}_2\mathbf{b}_1, \mathbf{c}_2 = \mathbf{P}_1^T\mathbf{c}_1$. Therefore,

$$\mathbf{H}_2 = \begin{bmatrix} \mathbf{H}_2^V \\ \mathbf{H}_2^W \end{bmatrix} = \begin{bmatrix} \mathbf{P}_1^T & 0 \\ 0 & \mathbf{P}_2 \end{bmatrix} \begin{bmatrix} \mathbf{H}_1^V \\ \mathbf{H}_1^W \end{bmatrix} = \hat{\mathbf{P}}^T\mathbf{H}_1.$$

According to the definition of graph isomorphism, $\mathcal{G}_1$ is isomorphic to $\mathcal{G}_2$.

### A.6 ALGORITHMS

---

**Algorithm 2** WL test for MILP/LP Graphs

---

**Require:** A graph instance $(G, H) \in \mathcal{G}_{m,n} \times \mathcal{H}_m^V \times \mathcal{H}_n^W$ and iterate limit $L > 0$.
1: Initialize with $C_i^{0,V} = HASH_{0,V}(h_i^V), C_j^{0,W} = HASH_{0,W}(h_j^W)$
2: **for** $l = 1, 2, \cdots, L$ **do**
3: $\quad C_i^{l,V} = HASH(C_i^{l-1,V}, \sum_{j=1}^n E_{i,j}HASH_{l,W}'(C_j)^{l-1,W})$
4: $\quad C_i^{l,W} = HASH(C_i^{l-1,W}, \sum_{j=1}^n E_{i,j}HASH_{l,V}'(C_j)^{l-1,V})$
5: **end for**
6: **return** The multisets containing all colors $\{\{C_i^{L,V}\}\}_{i=0}^m, \{\{C_i^{L,W}\}\}_{j=0}^n$.

---

---

**Algorithm 3** Determine if the graph is decomposable symmetric

---

**Require:** Graph $\mathcal{G}$'s adjacent matrix $\mathbf{A}$ and classification for stable partition[a] of it's variable nodes
$\quad \mathcal{I} = \{I_1, I_2, \cdots, I_J\}$.
1: Choose a index sets $I_i$ with $|I_i| > 1$.
2: **for** i : $I_i$ **do**
3: $\quad$ Search all constraint nodes that connect to node $i$ as $\mathcal{J}$
4: $\quad$ Exclude node $j \in \mathcal{J}$ if it is uniquely colored.
5: $\quad$ **for** j : $\mathcal{J}$ **do**
6: $\quad\quad$ Search all Variables that connect to node $j$ as $\mathcal{K}$
7: $\quad\quad$ **if** $\mathcal{K} \subseteq \mathcal{I}_1$ **then**
8: $\quad\quad\quad$ **pass**
9: $\quad\quad$ **else if** **then**
10: $\quad\quad\quad$ **return** False
11: $\quad\quad$ **end if**
12: $\quad$ **end for**
13: **end for**
14: **return** True

---

[a]See formal definition in Appendix A.8

## A.7 COMPLEXITY ANALYSIS

For the two main types of problem realizations in our benchmark, Algorithm 2 must converge in just one iteration. In addition, for problems with $m$ variables and $n$ constraints, the time complexity to distinguish tested problem realizations from the standard realization is at most $\mathcal{O}(mn + n \log n)$, which is is significantly lower than classical algorithms employed by popular solvers, such as simplex method for LP and branch and bound algorithm for MILP. Specifically,

1. **For WL-determinable problem instances**, Algorithm 2 converges after only 1 iteration, and the time complexity is $\mathcal{O}(mn)$.

2. **For decomposable symmetric problem instances**, Algorithm 2 converges after only 1 iteration, and we shall further conduct automorphism detection using algorithm 3, which takes time complexity $\mathcal{O}(n \ln n)$. The total time complexity could be $\mathcal{O}(mn + n \ln n)$

## A.8 PROOF FOR THEOREM 4.1

**Theorem A.1** *Denote Algorithm 1 by $\mathcal{A}(\mathcal{G}_{test}, \mathcal{G}_{standard})$. Suppose $\mathcal{P}_{standard}$ is WL-determinable or decomposable symmetric, then $\forall \mathcal{P}_{test}$, we have $\mathcal{A}(\mathcal{G}_{test}, \mathcal{G}_{standard}) ==$ True $\iff \mathcal{P}_{test} \sim \mathcal{P}_{standard}$.*

Before establishing the proof, we first introduce the coloring refinement process of WL test for MILP/LP problem since it is the first step 1 in algorithm $\mathcal{A}$. For iteration $l$ of the algorithm we will be assigning to each node a tuple $H_i^L$ containing the node's old compressed label and a multiset of the node's neighbors' compressed labels. A multiset is a set (a collection of elements where order is not important) where elements may appear multiple times.

At each iteration $l$, we will additionally be assigning to each node a new "compressed" label $C_i^L$ with the same $H_i^L$ will get the same compressed label.

Repeat the above process for up to (m+n) (the number of nodes) iterations or until the partition of nodes by compressed label does not change from one iteration to the next, we will get a converged multiset.

In addition, we introduce preliminary tools for an algorithm-independent definition.

In fact, WL-determinable and symmetric decomposable can be defined without relying on WL-test algorithm. We introduced equivalent definitions based on stable partition index sets.

**Definition A.3 (Stable Partition Index Sets)** *For a modeling instance $\mathcal{P}$ in the form of (1) with $n$ decision variables and $n$ constraints, define index set for optimization variables by $\mathcal{I} = \{I_1, I_2, \cdots, I_s\}$ and index set for constraints by $\mathcal{J} = \{J_1, J_2, \cdots, J_t\}$, where*

- $\bigcup_{l=1}^{s} I_l = \{1, 2, \cdots, m\}$, $\bigcup_{k=1}^{t} J_k = \{1, 2, \cdots, n\}$;

- $I_{l_i} \cap I_{l_j} = \emptyset$, $J_{k_p} \cap J_{k_q} = \emptyset$, $\forall i, j \in [1, \cdots, |I_l|], i \neq j$, and $p, q \in [1, \cdots, |J_k|], p \neq q$.

*The following condition holds:*

1. $(c_i, \tau_i) = (c_{i'}, \tau_{i'} \forall i, i' \in I_p$ for some $p \in 1, 2, \cdots, s$;

2. $(b_j, \circ_j) = (b_{j'}, \circ_{j'} \forall j, j' \in J_q$ for some $q \in 1, 2, \cdots, t$;

3. $\forall p \in 1, 2, \cdots, s, q \in 1, 2, \cdots, t$, and $i, i' \in I_p$, we have $\sum_{j \in J_q} a_{ij} = \sum_{j \in J_q} a_{i'j}$;

4. $\forall p \in 1, 2, \cdots, s, q \in 1, 2, \cdots, t$, and $j, j' \in J_q$, we have $\sum_{i \in I_p} a_{ij} = \sum_{i \in I_p} a_{ij'}$;

**Lemma A.2** *If there are no collision of hash functions and their weighted averages, then WL test will finally terminated at some stable partition.*

Lemma A.2 is proved in Chen et al. (2022b).

**Definition A.4 (WL-determinable, by trivial partition)** *$\mathcal{P}$ is WL-determinable if $\exists$ stable partition index sets $\mathcal{I}$ and $\mathcal{J}$ such that $\mathcal{I}$ or $\mathcal{J}$ are trivial partitions, i.e. $s = m$ and $t = n$.*

**Definition A.5 (Decomposable Symmetric, by grouped partition)** $\mathcal{P}$ *is decomposable symmetric if $\exists$ stable partition index set $\mathcal{I}$ and $\mathcal{J}$ such that:*

1. *There are only two types of index set in $\mathcal{I}$ and $\mathcal{J}$. Type 1 set only contains a single index. Type 2 contains several indexes, denote these sets by $I_1, \cdots, I_{s'}$; $J_1, \cdots, J_{t'}$.*

2. *$I_1, \cdots, I_{s'}$ and $J_1, \cdots, J_{t'}$ are equal-sized with $|I_p| = |J_q| > 1, \forall p \in \{1, 2, \cdots, s'\}$ and $q \in \{1, 2, \cdots, t'\}$.*

3. *$\forall p \in \{1, 2, \cdots, s'\}, q \in \{1, 2, \cdots, t'\}, i \in I_p, j \in J_q$, we have $|\{a_{ij}|a_{ij} \neq 0\}| = 1, \forall j \in J_q$ and $|\{a_{ij}|a_{ij} \neq 0\}| = 1, \forall i \in I_p$.*

By Lemma A.2, we can show two sets of definitions are equivalent.

Now we construct our proof by discussing two cases:

**Case 1:** Suppose $\mathcal{P}_{standard}$ is WL-determinable. Want to show $\mathcal{A}(\mathcal{G}_{test}, \mathcal{G}_{standard})$ == True $\iff \mathcal{P}_{test} \sim \mathcal{P}_{standard}$.

When $\mathcal{A}(\mathcal{G}_{test}, \mathcal{G}_{standard})$ == True and $\mathcal{P}_{standard}$ WL-determinable, we have $len(\mathbb{A}_1) = len(\mathcal{C}_1)$ & $len(\mathbb{A}_2) = len(\mathcal{C}_2)$.

By Algorithm 1, every color in the multisets output by WL test must be distinct and multisets for $\mathcal{P}_{standard}$ is the same as multisets for $\mathcal{P}_{standard}$. By Definition A.4, one stable partition of $\mathcal{G}_{standard}$ and is $\{I_1, \cdots, I_n\}, \{J_1, \cdots, J_m\}$, where $I_k, J_l$ are a single-element set, WLOG, assume $I_k = i_k, J_l = j_l$. Similarly, denote the stable partition of $\mathcal{G}_{test}$ by $\{I'_1, \cdots, I'_n\}, \{J'_1, \cdots, J'_m\}$, with $I'_k = [i_k], J'_l = [j'_l]$.

Now, define a bijection mapping that shuffles $[i_1, \cdots, i_m]$ and $[j_1, \cdots, j_n]$ to get $[i'_1, \cdots, i'_m]$ and $[j'_1, \cdots, j'_n]$, denote such mapping by $\mathbf{P}$. (Since each element in $[i_1, \cdots, i_m], [j_1, \cdots, j_n], [i'_1, \cdots, i'_m]$, or $[j'_1, \cdots, j'_n]$ is distinct, we can uniquely find such bijection).

Notice that such bijection may only map index of $v_i^{standard}$ to index of $v_j^{test}$, we can separately define a bijection for decision variable index as $\mathbf{P}_1$ and a bijection for constraint index as $\mathbf{P}_1$.

Therefore, exists bijection $\mathbf{P}_1$ and $\mathbf{P}_2$ such that $\mathcal{P}_{test}$ can be written in the following form:

$$\min_x \hat{c}^T x,$$
$$\text{s.t. } \hat{A}x \hat{\circ} \hat{b}, \hat{l} \leq x \leq \hat{u}$$

where $\hat{b} = P_2 b_{standard}, \hat{C} = P_1 C_{standard}, \hat{A} = P_2 A_{standard} P_1, \hat{\circ} = P_2 \circ_{standard}, \hat{l} = P_1 l_{standard}, u = P_1 u_{standard}$. This implies $\mathcal{P}_{test} \sim \mathcal{P}_{standard}$.

**Case 2** : Suppose $\mathcal{P}_{standard}$ is decomposible symmetric. When algorithm $\mathcal{A}$ output "Isomorphic", the partition sets of $\mathcal{G}_{standard}$ and $\mathcal{G}_{test}$ can be denoted as

$$\mathcal{I}_{standard} = [I_1, \cdots, I_k, I_{k+1}, \cdots, I_s];$$
$$\mathcal{J}_{standard} = [J_1, \cdots, J_l, J_{l+1}, \cdots, J_t];$$
$$\mathcal{I}_{test} = [\hat{I}_1, \cdots, \hat{I}_k, \hat{I}_{k+1}, \cdots, \hat{I}_s];$$
$$\mathcal{J}_{test} = [\hat{J}_1, \cdots, \hat{J}_l, \hat{J}_{l+1}, \cdots, \hat{J}_t],$$

where set $[I_1, \cdots, I_k], [\hat{I}_1, \cdots, \hat{I}_k], [J_1, \cdots, J_l], [\hat{J}_1, \cdots, \hat{J}_l]$, only contains one index, and set $[I_{k+1}, \cdots, I_s], [\hat{I}_{k+1}, \cdots, \hat{I}_s], [J_{k+1}, \cdots, J_t], [\hat{J}_{k+1}, \cdots, \hat{J}_t]$, are equal-sized and consisting at least 2 indexes.

By the definition of decomposable symmetric instances, for any two sets $K, S \in [I_{k+1}, \cdots, I_s, \hat{I}_{k+1}, \cdots, \hat{I}_s, J_{k+1}, \cdots, J_t, \hat{J}_{k+1}, \cdots, \hat{J}_t]$, $K$ and $S$ are either disconnected or exists a bijection connection between nodes from $K$ to $S$.

Now, define a bijection mapping that maps $[I_1, \cdots, I_k, I_{k+1}, \cdots, I_s, J_1, \cdots, J_l, J_{l+1}, \cdots, J_t]$ to $[\hat{I}_1, \cdots, \hat{I}_k, \hat{I}_{k+1}, \cdots, \hat{I}_s, \hat{J}_1, \cdots, \hat{J}_l, \hat{J}_{l+1}, \cdots, \hat{J}_t]$, by mapping one-element set to one-element set, and mapping multi-elements sets according to the connectivity between nodes in corresponding sets, denote such bijection by $\mathbb{P}$.

Similar to case 1, we can infer $\mathcal{P}_{test} \sim \mathcal{P}_{standard}$.

## A.9    Randomly sampling suffices to obtain WL-determinable instance

To make WL test work, it is desirable to sample a WL-determinable instance. In Theorem A.2, we proved that for a large range of modeling problems with **flexible property** (Definition A.7), especially for problems in our benchmark dataset, we can sample WL-determinable instance from its **parameter set** with probability 1.

**Definition A.6 (Modeling Parameter Set)** *For a class of model formulation $\mathcal{M}$ with $n$ decision variables and $m$ constraints, the **parameter set** $\mathcal{S}_{m,n}(\mathcal{M})$ is a collection of all possible values for problem data $(\mathbf{A}, \mathbf{c}, \mathbf{b}, \circ)$.*

An example of a formulation parameter set is attached in Appendix **??**.

Given a model's parameter set, we say model $\mathcal{M}$ is a flexible model if, for any variables $x_i$ in $\mathcal{M}$, at least one of its associated parameters —whether the objective coefficient or any of the constraint coefficients —can be arbitrarily chosen from its parameter set, which is expected to be sufficiently large. A formal definition of flexible model is as follows:

**Definition A.7 (Flexible Model)** *We say a model $\mathcal{M}$ is **flexible** if the following condition holds:*

*$\forall$ variables $x_i, i = 1, \cdots n, \exists$ element $p \in [\mathbf{A}_{:,i}^T, c_i]$ s.t. $p$ can be arbitrarily chosen from some uncountable set $S(p) \subset \mathbb{R}$. In other words, for given model $\mathcal{M}$, for any variables $x_i$ in $\mathcal{M}$, at least one of its associated parameters —whether the objective coefficient or any of the constraint coefficients —can be arbitrarily chosen from a sufficiently large space.*

**Theorem A.2 (Efficient Sampling)** *For a flexible model $\mathcal{M}$ with parameter set $\mathcal{S}_{m,n}(\mathcal{M})$, randomly sample in its parameter set under any continuous distribution may get a WL-determinable instance in probability 1.*

We present the proof for Theorem A.2 in Appendix A.10.

## A.10    Proof of Theorem A.2

**Proof:**    Claim 1: $P([\mathbf{A}_{:,i}^T, c_i] = \mathbf{A}_{:,j}^T, c_j]) = 0$ as long as $i \neq j$.

Suppose $\mathcal{M}$ is a flexible model. By the definition of the flexible model, for each variable $x_i$, there exists at least one element $p \in [\mathbf{A}_{:,i}^T, c_i]$ that can be randomly chosen from an uncountable set $S_p \subset \mathbb{R}$. This implies that the parameters corresponding to different indices $i$ and $j$ can vary independently within their respective uncountable set. Sampling from a continuous distribution over the parameter space $\mathcal{S}_{m,n}(\mathcal{M})$ involves independently sampling $[\mathbf{A}_{:,i}^T, c_i]$ from some continuous distribution for each $i$. Now, by the property of continuous sampling and independence of $[\mathbf{A}_{:,i}^T, c_i]$ and $[\mathbf{A}_{:,j}^T, c_j]$, we have

$$P([\mathbf{A}_{:,i}^T, c_i] = \mathbf{A}_{:,j}^T, c_j]) = 0,$$

i.e.

$$P([\mathbf{A}_{:,i}^T, c_i] \neq \mathbf{A}_{:,j}^T, c_j]) = 1.$$

Claim 2: Suppose a modeling instance $\mathcal{M}(\mathbf{s})$ has $[\mathbf{A}_{:,i}^T, c_i] \neq [\mathbf{A}_{:,j}^T, c_j], \forall i \neq j$, then this instance is WL-determinable.

Denote the index set that $\mathbf{e}_i = [\mathbf{A}_{:,i}^T, c_i]$ and $\mathbf{e}_j = [\mathbf{A}_{:,j}^T, c_j]$ differs by $K$, with $\forall k \in K, e_{ik} \neq e_{jk}$, $e_{ik}$ is the $k$-th element in vector $\mathbf{e}_i$.

It suffices to show that $\forall j \neq j'$, the joint probability of the following events is 1:

1. **Event A**: $c_j \neq c_{j'}$;

2. **Event B**: $\sum_{i \in I} a_{ij} \neq \sum_{i \in I} a_{ij'}$ for some $\mathcal{I}$;

3. **Event C**: $\sum_{q \in J} a_{i'q} \neq \sum_{q \in J} a_{iq}$ for some $J$ containing index $j$ or $j'$ and some $i \neq i' \in I$,

where $I, J$ elements in stable partition sets $\mathcal{I}, \mathcal{J}$.

Formally speaking, we want to show $P(A \cup B \cup C) = 1$. Now, consider two cases when $j \neq j' = 1, \cdots, n$.

Case 1: $e_{jk} = c_j, e_{j'k} = c_{j'}$ for some $k \in K$. Apparently, we have $c_j \neq c_{j'}$.

Case 2: $e_{jk} = a_{ij}, e_{j'k} = a_{ij'}$ for some $k \in K$ and some $i = 1, \cdots, m$. We have $a_{ij} \neq a_{ij'}$ for some $i = 1, \cdots, m$. We want to show $P(B \cup C | \text{Case 2}) = 1$.

Consider $I$ containing $i$, at least one element $i' \in I$ can be arbitrarily chosen from sufficiently large support and makes $a_{i'j} \neq a_{i'j'}$. WLOG, suppose $\hat{I} \subset I$ is a set containing all $i$'s such that $a_{ij} \neq a_{ij'}$, and for the remaining $i$'s, we have $\sum_{i \in I/\hat{I}}(a_{ij} - a_{ij'}) = c$, for some constant $c$.

$$P\left(\sum_{i \in I} a_{ij} \neq \sum_{i \in I} a_{ij'}\right) = P\left(\sum_{i \in \hat{I}} a_{ij} \neq \sum_{i \in \hat{I}} a_{ij'}\right)$$

$$= 1 - P\left(\sum_{i \in \hat{I}} a_{ij} - \sum_{i \in \hat{I}} a_{ij'} = -c\right)$$

$$= 1$$

The last equality holds since $\sum_{i \in \hat{I}} a_{ij'}$ and $\sum_{i \in \hat{I}} a_{ij}$ are independent and can be sampled from some continuous distribution.

$$P(A \cup B \cup C) = P(A \cup B \cup C | \text{Case 1} \cup \text{Case2})$$
$$= P(\text{Case 1})P(A \cup B \cup C | \text{Case 1}) + P(\text{Case 2})P(A \cup B \cup C | \text{Case2})$$
$$= P(\text{Case 1}) + P(\text{Case 2})$$
$$= 1$$

By theorem 2, we may get a WL-determinable instance in probability 1.

### A.11    EXAMPLES

**Example A.1 (Model Parameter Set for Blending Problem)** *For example, a blending problem can be formulated as:*

$$\min_x \sum_{i=1}^{n} c_i x_i$$

$$s.t. \sum_{i=1}^{n} a_{ji} x_i \geq p_j, \forall j = 1, \cdots, m.$$

$$x_i \leq u_i, \forall i = 1, \cdots, n.$$

*The corresponding parameter set $\mathcal{S}_{m,n}(\mathcal{M}_{blend})$ can be defined as*

$$\mathcal{S}_{m,n}(\mathcal{M}_{blend}) = \left\{ (\mathbf{A}, \mathbf{c}, \mathbf{b}, \circ) \middle| \mathbf{A} = [\hat{\mathbf{A}}^T, I_n]^T, \text{ where } \hat{\mathbf{A}} \in \mathbb{R}^{m \times n} \text{ and } I_n \text{ is an } n \times n \right.$$

$$\text{identity matrix}; \mathbf{c} = [c_1, \cdots, c_n]^T \in \mathbb{R}^n; \mathbf{b} = [-p_1, \cdots, -p_J, -u_1, \cdots, -u_n]^n \in \mathbb{R}^{m+n};$$

$$\left. \circ = [\geq, \cdots, \geq, \leq, \cdots, \cdots, \leq]_{1 \times (m+n)}^T \right\}.$$

*The parameter set associated with $x_i$ is $\mathcal{S}_{m,n}(\mathcal{M}_{blend}, i) = \left\{ [\mathbf{A}_{:,i}^T, c_i] \right\} = \mathbb{R}^{m+1}$.*

**Example A.2 (Undesirable Symmetry)** *Discriminating problem instances involving symmetry in their decision variables or constraints can be tricky. Because some non-isomorphic bipartite graphs cannot be distinguished by WL-test due to their automorphic structure in the graph. For example, Chen et al. (2022b) illustrates one case in which two MILP graphs are non-isomorphic while WL-test outputs the same multiset.*

$$\min_{x \in \mathbb{R}^6} x_1 + x_2 + x_3 + x_4 + x_5 + x_6,$$

$$\text{s.t. } x_1 + x_2 = 1, \ x_2 + x_3 = 1, \ x_3 + x_4 = 1,$$

$$x_4 + x_5 = 1, \ x_5 + x_6 = 1, \ x_6 + x_1 = 1,$$

$$0 \le x_j \le 1, \ x_j \in \mathbb{Z}, \ \forall \ j \in \{1, 2, \dots, 6\}.$$

$$\min_{x \in \mathbb{R}^6} x_1 + x_2 + x_3 + x_4 + x_5 + x_6,$$

$$\text{s.t. } x_1 + x_2 = 1, \ x_2 + x_3 = 1, \ x_3 + x_1 = 1,$$

$$x_4 + x_5 = 1, \ x_5 + x_6 = 1, \ x_6 + x_4 = 1,$$

$$0 \le x_j \le 1, \ x_j \in \mathbb{Z}, \ \forall \ j \in \{1, 2, \dots, 6\}.$$

Figure 9: Two non-isomorphic MILP graphs that cannot be distinguished by WL test

**Decomposable Symmetry Problem**   For decomposable symmetric problems, their corresponding bipartite graph can be divided into several symmetric sub-graphs, with each isomorphic and disconnected from others. For example, a instance on bin-packing with heterogeneous vehicles is formulated as

$$\min_{x \in \{0,1\}^q, y \in \{0,1\}^p} \sum_{j=1}^{p} y_j$$

$$\text{s.t. } \sum_i s_i x_{ij} \le b y_j, \forall j = 1, \cdots, p.$$

$$\sum_{j=1}^{p} x_{ij} = 1, \forall i = 1, \cdots, q$$

For the bin-packing problem with $p = 3$ and $q = 2$, a corresponding bipartite is illustrated in figure 10, where the red node represents decision variables and the blue nodes represent constraints.

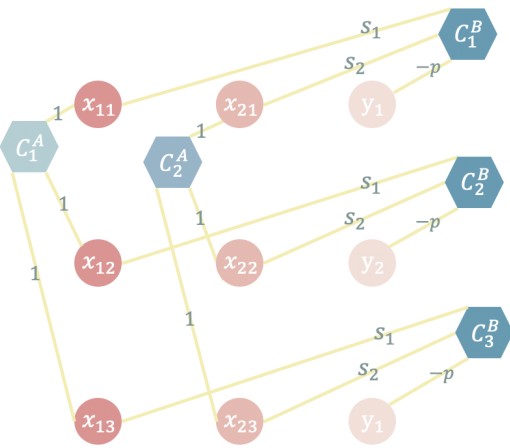

Figure 10: Bipartite for a bin-packing problem. Different colors indicate that the nodes are colored using the WL test.

This figure illustrates the representation of a symmetric decomposable graph. There are four groups of nodes with the same colors in each group, and two nodes with distinct colors. In addition, a node in any group, for example, the lightest red group, only connects with one node in other groups.

Such graphs are quite special since by excluding uniquely colored nodes and their connecting edges, the remaining symmetric nodes (nodes labeled in the same color via the WL test) can combined to form several isomorphic, disconnected, and WL-determinable graphs, as the dashed line highlights in Figure 11.

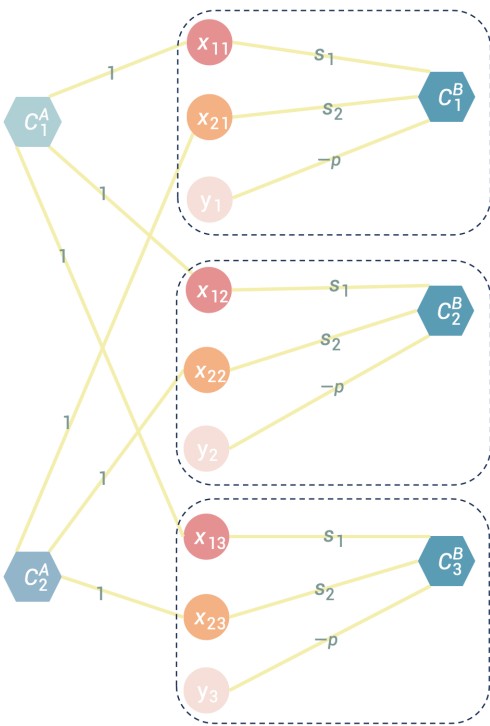

Figure 11: Decompose a decomposable symmetric graph

## A.12    ERROR ANALYSIS

We analyzed the source of the errors and observed that for most problems, the compilation errors in the generated code are relatively smaller than the modeling errors. This indicates that, in most cases, our benchmark assesses modeling capabilities rather than the LLMs' ability to generate solver code.

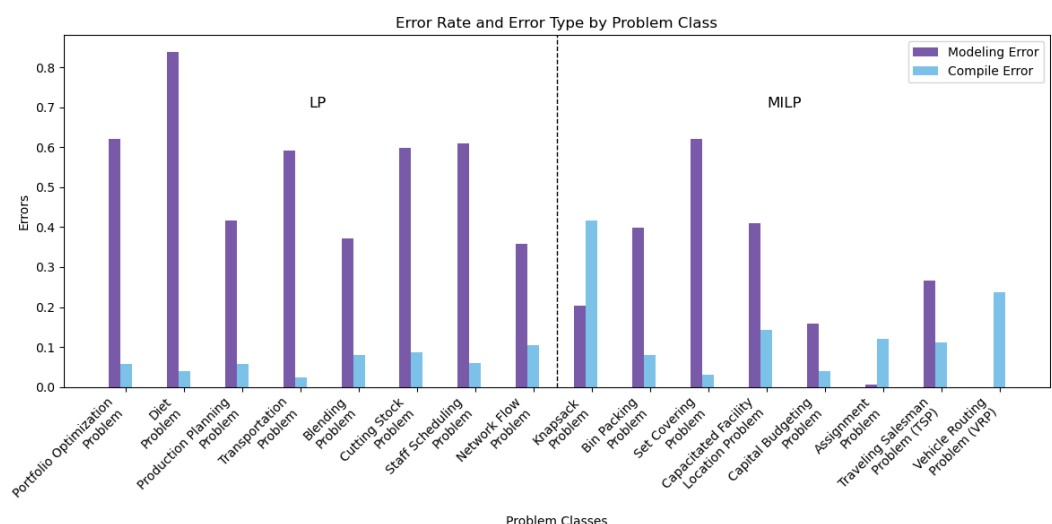

Figure 12: Modeling Error and Compiling Error for Different Problem Classes