# OpenReview forum: "OptiBench: Benchmarking Large Language Models in Optimization Modeling with Equivalence-Detection Evaluation"
_ICLR.cc/2025/Conference — ICLR 2025 Conference Withdrawn Submission_

### Official Review · Reviewer_qhCn · 2024-10-28

**Soundness:** 2
**Presentation:** 2
**Contribution:** 2
**Rating:** 5
**Confidence:** 2

**Summary:**

This paper introduces a benchmark dataset with 816 LP and MLP word problems. The authors propose a novel evaluation method based on graph theory for determining if two optimization problem instances are equivalent. Lastly, this work benchmarks recent LLMs and multi-agent approaches.

**Strengths:**

- A wide variety of CO problems: 16 classes, 816 problems.
- Presentation of each instance as a bipartite graph, along with a novel, efficient algorithm for detecting equivalence between instances.

**Weaknesses:**

- ChatGPT-4o is used to propose optimization models, create .json files, and translate solver code into a detailed world problem structure. This could introduce bias when benchmarking other LLMs.
- The only detailed instance described is the Knapsack Problem. I would be very interested in seeing other examples, especially for more complex problems, and supplementary material would be appreciated.

Minor remark: There are some technical errors in the paper, such as broken or missing references and unexpectedly inserted text (e.g., in line 244).

**Questions:**

- As I mentioned earlier, I feel that generating benchmark instances (model, data, and WP) with ChatGPT-4o makes them more easily understandable for ChatGPT-4o compared to other LLMs. This may explain why ChatGPT-4o performs the best on the benchmark dataset and why, "surprisingly, the o1-preview and o1-mini models underperformed compared to 4o". Have you considered this as a potential source of bias? I realize this is a significant amount of work, but it would be helpful to repeat the process using other LLMs in the data construction pipeline (Fig. 2), at least for some instances where ChatGPT-4o significantly outperforms other methods. This would be a way to eliminate potential bias in the benchmark toward ChatGPT-4o.

- I am not an expert in intellectual property, and I apologize if my question is trivial, but I’m curious about the licensing of the OptiBench dataset. Given that Gurobi is a commercial solver and is used in your pipeline, will your data be available for free? Additionally, if someone wants to extend your work, would they need a Gurobi license?

---

### Official Review · Reviewer_QcMr · 2024-11-04

**Soundness:** 2
**Presentation:** 4
**Contribution:** 3
**Rating:** 6
**Confidence:** 4

**Summary:**

The authors present a benchmark for evaluating the ability of large language models to formulate linear programs and mixed integer linear programs based on word problems and accompanying numerical data. The authors additionally propose to evaluate the quality of the generated optimization problem not by running a solver but instead by comparing the generated optimization problem to a ground truth optimization problem using graph isomorphism. The authors evaluate several language models on their benchmark, breaking down performance by type of optimization problem and identifying key limitations of existing approaches in several settings.

**Strengths:**

The paper itself is well written and the benchmark is promising in that existing methods don’t seem to solve all problems adequately, and exhibit a range of performance across the different types of problems, lending the benchmark to be used in different ways and to help understand how different models perform in settings with varying difficulty.

The diversity of problems investigated is representative of many textbook problems considered by the OR community, each with very different considerations in modeling.

The authors evaluate several approaches in the various settings, providing a breakdown of the results to help immediately identify areas for novel research.

**Weaknesses:**

It seems that since GPT-4o proposes the optimization models which are then filtered etc, the dataset might be biased towards problems that GPT-4o already has knowledge about. Is it possible that the dataset is not representative enough of realistic settings that GPT-4o is not performant on? It seems that might limit the applicability of the approach to represent realistic scenarios, especially those in which language-based modeling might fail due to lack of knowledge.

The graph isomorphism test is helpful, but is it possible that it misses reformulations which are equivalent but use different decision variables / constraints? For example, after presolve, the resulting problem is equivalent but has fewer decision variables and constraints. Alternatively, it may be that there are alternative formulations with additional constraints, such as specialized cuts, that improve the optimization model but which wouldn’t be considered correct by this evaluation.

Small comments:
-	The text in figure 1 is quite difficult to read as the font is small
-	Line 176: why is it “we leverage … “?
-	Line 243 has ??, the example of the word problem
-	Figure 2 has instanc

**Questions:**

Is there an idea of how efficient a given problem formulation is? For instance, with the right kinds of cuts or auxiliary variables one can greatly improve solver speed. It might be that two formulations are ultimately the same, but one can be a better formulation than the other.

You might consider citing specific works for the node selection in branching, or referencing survey papers about learning and optimization [1,2]

How does this collection of instances relate to the other datasets from previous work? Are there overlaps in the problem domains, or problems themselves?



[1] Bengio, Yoshua, Andrea Lodi, and Antoine Prouvost. "Machine learning for combinatorial optimization: a methodological tour d’horizon." European Journal of Operational Research 290.2 (2021): 405-421.
[2] Kotary, James, et al. "End-to-End Constrained Optimization Learning: A Survey." 30th International Joint Conference on Artificial Intelligence, IJCAI 2021. International Joint Conferences on Artificial Intelligence, 2021.

---

### Official Review · Reviewer_xRAt · 2024-11-05

**Soundness:** 4
**Presentation:** 2
**Contribution:** 3
**Rating:** 6
**Confidence:** 4

**Summary:**

The authors propose OptiBench, a benchmark to evaluate the ability of large language models (LLMs) for the modeling task of operation research (OR).  It provides 816 diverse problems across multiple types, classes, and domains. OptiBench separates model structure from data for higher scalability and introduces a theory-supported equivalence-checking approach using Weisfeiler-Lehman graph isomorphism tests for more precise model comparison. The authors claim that OptiBench provides new insights into the strengths and limitations of LLMs in optimization modeling.

**Strengths:**

1. Interesting evaluation metric. The authors propose an interesting and theory-supported equivalence-checking approach using Weisfeiler-Lehman graph isomorphism tests for more precise model comparison.
2. Comprehensive datasets. The paper provides a comprehensive range of data types, establishing a robust set of test benchmarks for mathematical modeling. This diversity greatly enhances the benchmark's applicability across different optimization scenarios, offering valuable resources for evaluating large language models in optimization tasks.

**Weaknesses:**

1. No datasets or results available. As a benchmark paper, it is essential to make the datasets and benchmark results publicly available to support reproducibility and usability—key criteria for evaluating the usability, effectiveness, and reliability of a benchmark.
2. Some writing issues, including undefined references (Line 243) and incorrect use of quotation marks (please use <`> and <'> in latex for quotation marks).

**Questions:**

1. Can this evaluation metric be applied to MILP problem generation [1] tasks?
2. Why is it necessary to separate data from the model? Please provide specific examples and illustrations to help a broader audience understand it.
3. Could you provide more insights about the selection of these datasets? Specifically, what aspects of LLM modeling capabilities do they each aim to test? Are there any key subsets of the whole dataset that could be used for authoritative and general evaluation?

[1] MILP-StuDio: MILP Instance Generation via Block Structure Decomposition.

---

### Official Review · Reviewer_eouH · 2024-11-11

**Soundness:** 3
**Presentation:** 2
**Contribution:** 2
**Rating:** 3
**Confidence:** 5

**Summary:**

The paper attempts to contribute along three axes:

1. A dataset of LP/MILP instances from a variety of motivating applications and combinatorial structures. The instances consist of a natural language description, numerical data, and an optimization model in the standard .lp format.

2. A graph isomorphism test to enable the evaluation of LLM-generated optimization formulations, rather than relying on exact matching which is not useful in this domain. The proposed test comes with a set of assumptions and corresponding correctness guarantees.

3. An empirical evaluation of a set of common LLM systems demonstrating superior performance for the gpt-4o model.

**Strengths:**

I will refer to the strengths by S1, S2, etc.

S1. I am impressed with the authors’ ambitious goal of covering a wide range of optimization applications. I have questions and concerns about how those applications and formulations were derived, by I applaud the attempt at building a comprehensive database.

S2. The algorithmic/theoretical contribution to testing equivalence between LPs/MILPs is very interesting. Evaluating LLM performance in constructing optimization formulations from natural language necessitates a reliable automated testing procedure. Posing this question in the paper is a strength.

S3. The empirical evaluation brings about some interesting findings as to which LLMs are better at this task. Many operations researchers and educators are curious about this question at the moment.

S4. The positioning of the paper relative to prior work is excellent. This is despite many concurrent papers in this fast-evolving space.

**Weaknesses:**

W1. Writing: I include some detailed comments in the Questions box. In summary, the writing and presentation quality has a ways to go and should be improved before this work is published.

W2. Lack of details on OptiBench problems: The reader is referred to appendix Table 2 for further details on the problems and applications in this benchmark. However, Table 2 only has eight LP types and eight MILP types. These are all standard textbook optimization problems. While appendix Figure 6 features a cargo loading application, it is unclear what other applications are in this benchmark. As a reader, I am left wondering how complex your problems are from a modeling perspective. This makes it impossible to assess whether LLM performance is meaningful for real-world deployment or not.

W3. Use of LLMs to generate the problems: I know that “generative AI” research heavily uses generative models such as LLMs to generate synthetic data for benchmarking purposes. However, in the operations research setting, this seems unnecessary given that the use of LLMs introduces risks of hallucination and other side effects. The authors claim that a quality control process was established using expert operations researchers that went through every LLM-generated problem and discarded spurious ones; there is not enough detail about this process to be able to trust the benchmark’s quality.

Here are 48 distinct, well-defined, well-motivated variants of TSP/VRP, for example: http://webhotel4.ruc.dk/~keld/research/LKH-3/. You can simply generate instances of these problems along with the corresponding natural language description; the papers that introduced these problems also contain further context that you can leverage. The same can be said of scheduling, for example [http://www.schedulingbenchmarks.org/] or maritime inventory routing [https://mirplib.scl.gatech.edu/home] or data center-type problems [see problems 1 and 2 in Datasets at https://www.ecole.ai/2021/ml4co-competition/].

W4. Generality of the proposed equivalence test: It seems that an important aspect of formulation equivalence is ignored in your formal definition A.1. Namely, from a strictly logical standpoint, two optimization models are said to be equivalent if their feasible regions are the same and the objective function values at feasible points are equal. For MILP, this is extremely crucial as for the same problem some formulation may be “weak” whereas another may be “strong”; this typically relates to the LP relaxation’s tightness. The textbook example for this is in Facility location: https://scipbook.readthedocs.io/en/latest/flp.html#weak-and-strong-formulations. Defining equivalence in terms of the variable-constraint graph is clearly limiting in this respect and needs to be addressed explicitly. If your test cannot handle it, that should be made clear. I believe that is the case. It is quite likely that any test for this problem will struggle with this issue.

**Questions:**

Kindly respond to the four weaknesses above. Minor comments to improve the quality of presentation are provided below.

Minor comments:
- “word problems” in abstract: might be better to refer to them as “text” problems or “natural language” problems.
- Figure 1 and others: The captions should be expanded to explain the figure in a self-contained way. The reader should not have to search in the body of the paper to understand Figure 1, for example.
- Page 3, 142: MLLP -> MILP
- Eq. (1): suggestion: use $^\intercal$ for the transpose instead of $^T$.
- Eq. (1): I am surprised strict inequality constraints are allowed. These cannot be handled by LP/MILP solvers.
- Page 5, 217: please add spaces before parentheses e.g., “cutting stock (LP)” instead of “cutting stock(LP)”. Also applies to citations and references to Tables/figures, e.g., “Table2” on line 218.
- Page 5, 243: broken reference “??”. Same in appendix line 1151 and possible others
- Page 5, 245: “being prohibitive” —> “being restricted by the LLM’s…”
- Figure 2: “Instanc” —> “Instance”
- Page 7, 337: “graph isomorphic” —> “graph isomorphism”
- Page 9, 473: open parentheses not closed! “(We monitor…”

---

### Note · Authors · 2024-11-24

I have read and agree with the venue's withdrawal policy on behalf of myself and my co-authors.